# Intelligent Fault Diagnosis Method Based on VMD-Hilbert Spectrum and ShuffleNet-V2: Application to the Gears in a Mine Scraper Conveyor Gearbox

**DOI:** 10.3390/s23104951

**Published:** 2023-05-21

**Authors:** Weibing Wang, Shuai Guo, Shuanfeng Zhao, Zhengxiong Lu, Zhizhong Xing, Zelin Jing, Zheng Wei, Yuan Wang

**Affiliations:** School of Mechanical Engineering, Xi’an University of Science and Technology, Xi’an 710054, China; 20205224126@stu.xust.edu.cn (S.G.); zsf@xust.edu.cn (S.Z.); 17868879706@163.com (Z.L.); zzxing64@163.com (Z.X.); 13259716754@163.com (Z.J.); 20205224114@stu.xust.edu.cu (Z.W.); wangy1212@xust.edu.cn (Y.W.)

**Keywords:** MCSA, load impact, VMD, GA, Hilbert spectrum, ShuffleNet-V2

## Abstract

This paper introduces a fault diagnosis method for mine scraper conveyor gearbox gears using motor current signature analysis (MCSA). This approach solves problems related to gear fault characteristics that are affected by coal flow load and power frequency, which are difficult to extract efficiently. A fault diagnosis method is proposed based on variational mode decomposition (VMD)–Hilbert spectrum and ShuffleNet-V2. Firstly, the gear current signal is decomposed into a series of intrinsic mode functions (IMF) by using VMD, and the sensitive parameters of VMD are optimized by using a genetic algorithm (GA). The Sensitive IMF algorithm judges the modal function sensitive to fault information after VMD processing. By analyzing the local Hilbert instantaneous energy spectrum for fault-sensitive IMF, an accurate expression of signal energy changing with time is obtained to generate the local Hilbert immediate energy spectrum dataset of different fault gears. Finally, ShuffleNet-V2 is used to identify the gear fault state. The experimental results show that the accuracy of the ShuffleNet-V2 neural network is 91.66% after 778 s.

## 1. Introduction

With the increase in global demand for energy, coal mining equipment is developing in the direction of being large-scale and intelligent [1,2]. Scraper conveyors are mainly responsible for conveying coal in coal mining. However, due to the complex and harsh mine production environment, it is difficult to intelligently monitor the state of the scraper conveyor. The transmission gear fault accounts for a significant probability of scraper conveyor failure [3,4], so it is necessary to study the fault diagnosis of the scraper conveyor gearbox.

Researchers have made significant contributions to the fault diagnosis of gearboxes, among which the vibration signal is the most common condition detection method [5,6]. However, the scraper conveyor in the process of coal transportation, the coal drop will cause a certain intensity of impact extrusion to the vibration sensor, resulting in the vibration sensor loose, falling off, or even being damaged; it is difficult to use effectively for a long time. The method based on motor current signature analysis (MCSA) is called the most promising method of equipment condition monitoring. MCSA is based on the fact that the gear fault will produce torque oscillation [7], and the stator current of the motor will change regularly due to mechanical–magneto-electric interaction, so the gear state can be monitored by analyzing the current signal [8,9]. The in-depth study of MCSA has been used in the field of motor diagnosis and monitoring and diagnosis of motor transmission chains. For example, Mohanty et al. applied MCSA to the state detection of multistage gearboxes and used the demodulated current signal to detect gear faults. The results show that the main component of the current signal is the power frequency signal and the frequency change caused by the defect is relatively weak [10].

However, using MCSA to monitor the condition of the mine scraper conveyor gearbox faces two complex problems. First, for the coal transport process in the mine, the scraper chain has to overcome great friction resistance to run in the chute. With the coal cutter moving to cut coal, the load change in the scraper conveyor has time-varying and random characteristics. In addition, the electrical system will introduce substantial power frequency interference, which eventually leads to the current signal of the load motor mainly representing power frequency signal and load impact information. In contrast, the characteristic information caused by the fault gear is challenging to extract [11].

The signal processing methods based on feature extraction play an essential role in the field of gear fault diagnosis, including wavelet transform (WT), empirical mode decomposition (EMD), variational mode decomposition (VMD), among others. For example, Xiao et al. use wavelet packet transform (WPT) to decompose and reconstruct the signal, calculate the energy value of each component, take the energy value as the feature input, and finally realize the fault diagnosis of the gearbox [12]. In addition, EMD can adaptively decompose the original signal into several IMF from high frequency to low frequency [13], so EMD can reduce the influence of unrelated components. For example, Han et al. decomposes nonstationary signals into several IMF by EMD, and selects IMF with sensitive fault characteristic frequency as the input of SVM for gear fault diagnosis [14]. According to the operating parameters and frequency spectrum, Muhammad et al. selected IMF, including gear fault characteristics, and finally realized the fault diagnosis of gearbox gears [15]. In addition, VMD is widely used. Through noniterative decomposition and adaptive frequency band selection, VMD decomposes complex original signals into different IMFs reflecting the dynamic mechanism of mechanical systems and extract’s fault features by analyzing the modal information where faults are located. For example, Liu et al. directly extract global fault features using VMD and combining SVD and obtain a total recognition rate of 100% by combining convolution neural networks [16]. Fan et al. established the dynamic model of the gearbox under unstable heavy load conditions. They obtained the fault frequency caused by the tooth root crack of the gearbox by numerical simulation. The VMD algorithm is used to prove the specific spectrum expression of the faulty gear [17]. Zhang et al. improved the fault diagnosis accuracy of the gearbox under variable load by using the VMD–sample entropy eigenvector construction method. They optimized the sensitive parameters of VMD by using the grasshopper optimization algorithm (GOA) and finally realized high-precision fault diagnosis under variable load conditions [18].

Although these signal processing methods have advantages, they need prior knowledge for feature extraction and rely heavily on expert systems. For example, in the process of WPT, the basis function needs to be preselected according to the characteristic information of the fault gear of the scraper conveyor, which limits the application of WPT. It should not be ignored that EMD is prone to mode aliasing and endpoint effect, which may affect the results of fault feature extraction. Furthermore, VMD has the problem that it is difficult to determine the sensitive parameters for the decomposition process.

However, recently developed machine learning technology avoids separating fault features from complex signals. Deep learning can find features from original data through multilayer nonlinear data processing units, and has achieved great success in gear fault diagnosis [19]. For example, He et al. extracted the original bearing vibration signal by wavelet packet transform, obtained the spectrogram, and then sent it to a convolution neural network to adjust the parameters by the simulated annealing algorithm. Finally, the accuracy in diagnosing different fault types was 97% [20]. Zhou constructed a one-dimensional residual depth neural network for gearbox fault identification, which significantly improved the accuracy of diagnosis [21]. Moslem realizes the fault diagnosis of the motor gearbox by multisensor fusion based on a two-dimensional convolution network and the current signal [22]. Although the fault diagnosis of gearboxes based on neural networks has achieved good results, with increased data complexity, neural networks need more complex structure, and the training process usually consumes much time and requires more computing resources. Therefore, Ma and others put forward the ShuffleNet-V2 model, introducing channel split and channel shuffle operations, which improves the running speed of the model and reduces the computing resources consumed by the training network. The ShuffleNet model is also widely used in fault pattern recognition because of its good balance between computational complexity and accuracy and has achieved good results [23]. For example, Luo et al. used complementary integrated empirical mode decomposition to denoise the signal, followed by conversion of the denoised signal into an angular domain signal and obtained the envelope spectrum of the angular domain signal by Hilbert transform. Finally, ShuffleNet-V2 is used to classify different fault types. Experimental results show that the proposed method still has high training accuracy without significantly increasing the model size [24].

To solve these problems, this article combines the advantages of signal processing technology and a deep network model in data preprocessing and feature learning. It proposes a transmission gear fault diagnosis model based on the ShuffleNet-V2 network and VMD-GA-Hilbert spectrum. The proposed model can be divided into three stages. The first stage is preprocessing from the original input signal. In the second stage, Hilbert’s instantaneous energy spectrum is used to express the fault gear characteristic information of the scraper conveyor under load impact. The third stage is to classify different types of fault modes by using a neural network. To extract useful fault feature information from the original data, every 10,000 data points of the input signal are segmented, and the original signal is decomposed into a series of IMF sub-band signals by VMD. The sensitive parameters involved in the decomposition process of VMD are determined by using local envelope entropy as the adaptability function of GA. Additionally, the fault sensitivity discrimination algorithm is used to extract the sub-band signals representing the fault. Secondly, the local Hilbert instantaneous energy spectrum expresses the time, frequency, and energy of the fault-sensitive signals to generate color-feature image datasets of different fault types. It is then used as the input data for the ShuffleNet-V2 neural network. Finally, the neural network can learn the characteristics of faulty gears and classify them under interference and their structure, as shown in Figure 1.

This research also compares the proposed fault diagnosis method for the scraper conveyor gearbox with other fault diagnosis models, including the generation method for the fault-feature dataset of wavelet time–frequency images. It uses a ResNet-18 neural network to classify different fault modes. Finally, the effectiveness of the proposed fault diagnosis method applied to mine scraper conveyor gearbox fault diagnosis is verified by comparative experiments.

## 2. Methodology

### 2.1. VMD

The VMD method is essentially an adaptive Wiener filter bank with narrow-band characteristics. The corresponding modal components $u_{k}$ are not fixed in the frequency domain but change with the change in the current signal. The ultimate goal is to transfer the input current signal $i_{ia}$ decomposed into discrete forms of different modes [25,26]. VMD operation is mainly divided into the construction of variational problems and the solution of variational problems [27,28]:

#### 2.1.1. Construction of Variational Problems

Step 1: The signal $i_{ia}$ is decomposed into the sum of K functions $u_{k}(t)$, as in Equation (1):(1)$$i_{ia}=\sum_{k=1}^{K}u_{k}(t)$$
where $u_{k}(t)$ is the decomposition to obtain *K* IMF components.

Step 2: The Hilbert transformation is performed on each mode function $u_{k}(t)$ to obtain the transformed analytic signal, and the unilateral spectrum can be solved:(2)$$\left[\delta \left(t\right)+\frac{j}{\pi t}\right]*u_{k}(t)$$

Step 3: The center frequency of each mode analytical signal is estimated, and the frequency spectrum of each mode is modulated and distributed to the corresponding fundamental frequency band:(3)$$\left[\left(\delta \left(t\right)+\frac{j}{\pi t}\right)*u_{k}(t)\right]e^{-j\omega_{k}t}$$
where $\left\{\omega_{k}\right\}$ are the respective center frequencies of the *K* IMF components.

Step 4: The bandwidth of each mode function is estimated; then, the corresponding constraint variational problem model is as follows:(4)$$\left\{\begin{array}{c} {min}_{\left\{u_{x}\right\}\left\{\omega_{x}\right\}}\left\{\sum_{k}{\left\|\partial_{t}\left[\left(\delta \left(t\right)+\frac{j}{\pi t}\right)*u_{k}\left(t\right)\right]e^{-j\omega_{k}t}\right\|}^{2}\right.\\ s.t\sum_{k}u_{k}=i_{ia} \end{array}\right.$$

In the formula, δ(t) is the pulse signal, $u_{k}$ is the decomposition to obtain *K* modal components, and $\omega_{k}$ represents the central frequency of the bandwidth.

#### 2.1.2. Solution of Variational Problems

To ensure the absolute integrability of the signal, a quadratic penalty factor α and Lagrange multiplication operator λ(t) are introduced, which can guarantee the accuracy of the reconstructed signal and the rigor of the constraint conditions, respectively. The constrained variational problem of Formula (4) is finally transformed into an unconstrained problem, such as Formula (5) [29]:(5)$$\begin{array}{c} L\left(\left\{u_{k}\right\},\left\{\omega_{k}\right\},\lambda \left(t\right)\right)=\alpha \sum_{k}{\left\|\partial_{t}\left[\left(\delta \left(t\right)+\frac{j}{\pi t}\right)*u_{k}\left(t\right)\right]e^{-j\omega_{k}t}\right\|}_{2}^{2}\\ \;+{\left\|f\left(t\right)-\sum_{k}u_{k}\left(t\right)\right\|}_{2}^{2}+\langle \lambda \left(t\right),f\left(t\right)-\sum_{k}u_{k}\left(t\right)\rangle \end{array}$$
wherein $u_{k}^{n+1}$, $\omega_{k}^{n+1}$, and $\lambda^{n+1}$ are updated alternately by alternating directions of multiplication operators. $u_{k}^{n+1}$ can be calculated as follows:(6)$$u_{k}^{n+1}\left(t\right)=argmin\left\{{\alpha \left\|\partial_{t}\left[\left(\delta \left(t\right)+\frac{j}{\pi t}\right)*u_{k}\left(t\right)\right]e^{-j\omega_{k}t}\right\|}_{2}^{2}+{\left\|f\left(t\right)-\sum_{i\neq k}^{K}u_{i}\left(t\right)+\frac{\lambda (t)}{2}\right\|}_{2}^{2}\right\}$$

Formula (6) is transformed into the frequency domain:(7)$$\begin{array}{c} {\hat{u}}_{k}^{n+1}\left(t\right)=argmin\{{\alpha \left\|j\omega \left[\left(1+sgn\left(\omega +\omega_{k}\right)\right)*\hat{u}\left(\omega +\omega_{k}\right)\right]\right\|}_{2}^{2}\\ \;+{\left\|\hat{f}\left(\omega \right)-\sum_{i}{\hat{u}}_{i}\left(\omega \right)+\frac{\hat{\lambda }(\omega )}{2}\right\|}_{2}^{2}\} \end{array}$$

Thus, the updating method of ${\hat{u}}_{k}^{n+1}$ is as follows:(8)$${\hat{u}}_{k}^{n+1}=\frac{\hat{f}\left(\omega \right)-\sum_{i\neq k}{\hat{u}}_{i}\left(\omega \right)+(\hat{\lambda }(\omega )/2)}{{1+2\alpha (\omega -\omega_{k})}^{2}}$$

Similarly, the updating methods of $\omega_{k}^{n+1}$ and $\lambda^{n+1}$ can be obtained:(9)$$\omega_{k}^{n+1}=\frac{\int_{0}^{\infty }\omega {\left|{\hat{u}}_{k}^{n}(\omega )\right|}^{2}d\omega }{\int_{0}^{\infty }{\left|{\hat{u}}_{k}^{n}(\omega )\right|}^{2}d\omega }$$

According to the above theory, the VMD calculation flow is as follows:

Step 1: Initialize the values of $\left\{u_{k}\right\}$, $\left\{\omega_{k}\right\},\lambda$, and i;

Step 2: Execute i=i+1, cyclic increment program;

Step 3: Update $u_{k}$, $\left\{\omega_{k}\right\}$ based on Equations (8) and (9);

Step 4: Execute k=k+1, repeat step 3 until k=K;

Step 5: Update λ according to Formula (10):(10)$${\hat{\lambda }}^{n+1}={\hat{\lambda }}^{n}\left(\omega \right)+\tau \left(f\left(\omega \right)-\sum_{k}{\hat{u}}_{k}^{n+1}\left(\omega \right)\right)$$

In this form, ∧ is the frequency domain transform and n is the number of iterations;

Step 6: Set the discrimination accuracy ε until the iteration stop condition such as Formula (11) is satisfied, the loop is ended, and the K modal component output.
(11)$$\sum_{K}{\left\|{\hat{u}}_{k}^{n+1}-{\hat{u}}_{k}^{n}\right\|}_{2}^{2}∕{\left\|{\hat{u}}_{k}^{n}\right\|}_{2}^{2}<\varepsilon$$

### 2.2. GA Optimizes VMD Parameters

Using this analysis, VMD faces the parameter selection problem. To make the fault signal of the scraper conveyor match the best decomposition effect to the maximum extent, it is necessary to optimize the value of α, K [30]. GA is a global random search method proposed by Professor Holland according to the evolution mechanism of natural species. In this algorithm, the code string formed by the parameters to be optimized in VMD is simulated as a biological evolution process. The next generation is generated by simulating natural genes’ crossover and mutation operations using the probability optimization method, so the search direction and space are automatically adjusted. The fitness value of individuals in the group is continuously improved until a certain termination condition is met. The steps of GA optimization VMD parameters are as follows [31,32]:

(1) Initialization: Set the evolution algebra counter t=0, set the maximum evolution algebra T, and randomly generate M individuals as the initial population P(0).

(2) Individual evaluation: Calculate the fitness of each individual in the group P(t).

(3) Select operation: Apply the selection operator to the group. The purpose of the selection operation is to pass the optimized individual to the next generation directly or to generate a new individual through pairing to regenerate to the next generation. The selection operation is based on the fitness assessment of the individual in the group.

(4) Crossover operation: Apply crossover operators to groups. The core function of the genetic algorithm is the crossover operator. For crossover operation, its primary purpose is to create more excellent offspring and improve the population’s adaptability and optimization level. First, two individuals are randomly selected from the population, and through the exchange and combination of two chromosomes, the excellent characteristics of the parent string are inherited by the substring, thus generating new excellent individuals. Assuming that $x_{a}\left(n\right)$ and $x_{\beta }(n)$ are the nth genes of the parent chromosome, the nth genes $x_{a}^{\prime }(n)$ and $x_{\beta }^{\prime }(n)$ of the offspring after crossover are shown in Formula (12), respectively.
(12)$$\left\{\begin{array}{c} x_{a}^{\prime }\left(n\right)=\left(1-\beta \right)x_{a}\left(n\right)+\beta x_{\beta }(n)\\ x_{\beta }^{\prime }\left(n\right)=\beta x_{a}\left(n\right)+\left(1-\beta \right)x_{\beta }(n) \end{array}\right.$$
where β is a random number between {0, 1}.

(5) Mutation operation: The mutation operator is applied to a group that randomly mutates an individual’s genes to generate a new one. Mutating individuals in the search process is necessary to jump out of the local minimum and accelerate the algorithm’s convergence. Choose a smaller value as the variation rate, and randomly select one of some individual genes in the new population for inversion. Suppose a new individual is produced by mutating the nth gene x(n) on the chromosome, and the mutation result is shown in Equation (13).
(13)$$x^{\prime }\left(n+1\right)=\left\{\begin{array}{c} x\left(n\right)+\left|\varepsilon_{1}\left\{max\left[x\left(n\right)\right]-x(n)\right\}\right|\varepsilon_{1}<1/2\\ x\left(n\right)+\left|\varepsilon_{2}\left\{x(n)-min[x(n)]\right\}\right|\varepsilon_{2}\geq 1/2 \end{array}\right.$$
where $\varepsilon_{1},\varepsilon_{2}\in (0,1)$.

(6) Termination condition judgment: If T, the calculation is terminated by using the individual with the greatest fitness obtained in the evolution process as the optimal solution output [33].

In the process of the coal conveyor, in addition to the fault signal, the frequency information is mainly caused by a large number of random load impacts in the motor current signal. The pulsating effect caused by the fault gear is different from other random impact signals, which will show certain periodicity and regularity, making the characteristic fault signal have strong sparsity, and the corresponding information entropy value is relatively small. The important aspect is that the local envelope entropy can effectively reflect the irregularity and complexity of the signal, and the difference in its value corresponds to the degree of uncertainty in the signal. The greater the information entropy value, the greater the corresponding uncertainty [34,35]. The envelope signal obtained from the VMD demodulation operation is processed into a probability distribution sequence $P_{i}$. The entropy value calculated by it reflects the sparse characteristics of the original signal, and the envelope entropy of the zero mean signal x(j)(j=1,2,⋯,N) is expressed by $E_{P}$, such as the Formula (14):(14)$$\left\{\begin{array}{c} E_{p}=-\sum_{j=1}^{N}p_{j}lgp_{j}\\ e_{j}=a\left(j\right)∕\sum_{j=1}^{N}a\left(j\right) \end{array}\right.$$In the equation, $P_{i}$ is the normalized form of $a_{i}$, and $a_{i}$ is the envelope signal obtained by Hilbert demodulation of signal $x_{j}$.

### 2.3. Sensitive IMF Discriminant Algorithm and Hilbert Transform

The signals are decomposed by VMD-GA to obtain a series of IMF. Although VMD can effectively extract the signals representing fault sub-bands and reduce the generation of false IMF components, for mechanical fault diagnosis, irrelevant IMF will affect fault diagnosis accuracy for mechanical fault diagnosis. At the same time, only some IMF in real IMF contains fault features or is sensitive to fault features. Therefore, before the Hilbert transform of IMF, it is necessary to judge the correlation degree between each IMF component and the diagnosed fault, so as to ensure the accuracy of fault feature extraction and the effectiveness of diagnosis [36]. In this paper, a sensitive IMF judgment method based on signal correlation is adopted. The specific algorithm is as follows:

Step 1: Calculate the correlation coefficient $c_{1}\left(t\right),\ldots ,c_{n}\left(t\right)$ between each IMF component of the fault signal x(t) and the original signal $\alpha_{i}$;

Step 2: Calculate the IMF components of the fault signal x(t) and the correlation coefficient $\beta_{i}$ between $c_{1}\left(t\right),\ldots ,c_{n}\left(t\right)$ and normal state signal $y\left(t\right)$.

Step 3: The sensitivity correlation coefficient $\gamma_{i}$ of each IMF component containing fault information is calculated from two correlation coefficients, such as Formula (15):(15)$$\gamma_{i}=\left|\alpha_{i}-\beta_{i}\right|(i=1,2,\ldots n)$$

Step 4: The fault sensitivity coefficient of each IMF is calculated by the Formula (16):(16)$$\lambda_{i}=\frac{\gamma_{i}-min(\gamma )}{\max\left(\gamma \right)-min(\gamma )}$$

Step 5: The IMF is reordered according to the sensitivity coefficient to obtain a new IMF sequence $\left\{c_{n}^{\prime }\right\}(i=1,2,\ldots n)$, and the difference between the two adjacent sensitivity coefficients is calculated by Equation (17):(17)$$d_{n}=\lambda_{n}^{\prime }-\lambda_{n-1}^{\prime }$$

Step 6: Determine the sequence number *K* corresponding to the maximum difference of sensitivity coefficient, then the first *K* IMF after sorting is the fault sensitive IMF.

Then, the Hilbert transformation of the extracted IMF components representing the fault signal can obtain the instantaneous amplitude and instantaneous frequency. For the original data, ${\overset{-}{c}}_{k}$ is decomposed as in Equation (18):(18)$${\overset{-}{c}}_{k}=\frac{1}{\pi }\int_{-\infty }^{\infty }\frac{c_{k}\left(\tau \right)}{t-\tau }d\tau$$

Construct the analytical signal by Equation (19),
(19)$$z\left(t\right)=c_{k}\left(\tau \right)+j\bar{c_{k}}\left(t\right)=a_{k}\left(t\right)e^{j\theta_{k}\left(t\right)}$$
where
(20)$$a_{k}\left(t\right)=\sqrt{{c_{k}(t)}^{2}+\overset{-}{c_{k}}{\left(t\right)}^{2}}$$
(21)$$\theta_{k}\left(t\right)=arctan\left(\frac{\overset{-}{c_{k}}\left(t\right)}{c_{k}\left(t\right)}\right)$$In Equations (20) and (21), $a_{k}\left(t\right)$ represents the instantaneous amplitude of $c_{k}\left(\tau \right)$ and $\theta_{k}\left(t\right)$ represents the instantaneous phase of $c_{k}\left(\tau \right)$.

The instantaneous frequency can be obtained by Equation (22),
(22)$$\omega_{k}\left(t\right)=\left(\frac{d\theta_{k}(t)}{dt}\right)$$

The main principle of this method is that the greater the correlation coefficient between the fault signal *x* (*t*) and the IMF component, the more fault information the IMF component contains. By calculating the correlation coefficient $\beta_{i}$ with the normal state signal y(t), we can know the normal information in each IMF component that is independent of the fault information [37]. Therefore, before the Hilbert transform of IMF, it is necessary to judge the correlation degree between each IMF component and the diagnosed fault to ensure the accuracy of fault feature extraction and the effectiveness of diagnosis. The sensitive IMF discrimination method combines the correlation between each IMF and fault signal with the correlation of the normal signal, and the combination of Hilbert transform can highlight the fault information and weaken the influence of the normal information [38].

### 2.4. Local Hilbert Instantaneous Energy Spectrum

Combining the instantaneous frequency and amplitude of this sensitive IMF, the local Hilbert spectrum of the signal can be obtained, as shown in Equation (23):(23)$$H^{\prime }\left(\omega ,t\right)=Re\sum_{k=1}^{n}a_{k}(t)exp\left(j\int \omega_{k}(t)dt\right)$$In the formula, Re is the real part, and $H\left(\omega ,t\right)$ accurately describes the variation in the amplitude (energy) of the signal with time and frequency. Therefore, if ${\left|x(t)\right|}^{2}$ is regarded as the energy density of the signal, the Hilbert spectrum has the same physical meaning after HHT analysis, and $H^{2}\left(\omega ,t\right)$ is called Hilbert energy spectrum. According to the theory of conservation of energy, Formula (24) holds:(24)$$\int_{-\infty }^{\infty }{\left|x(t)\right|}^{2}dt=\int_{-\infty }^{\infty }\int_{-\infty }^{\infty }H^{2}\left(\omega ,t\right)d\omega dt$$It can be defined from this:(25)$$E^{\prime }\left(t\right)=\int_{-\infty }^{\infty }H^{\prime 2}(\omega ,t)d\omega$$Equation (25) is the local Hilbert instantaneous energy spectrum, which reflects the distribution of signal energy with time [39].

## 3. Experiments

### 3.1. Current Data Acquisition

This section introduces the research on the transmission gear of a mine scraper conveyor with the actual coal mining process as the research background. Because the scraper conveyor belongs to low-speed and heavy-duty machinery, its working current often reaches several hundred amperes. So, it is necessary to realize current signal conversion through the current transformer and then collect the converted three-phase current through a 24-bit acquisition card. The selected current sampling frequency is 10 KHz, and the data acquisition process is shown in Figure 2.

HB-KPL-75 mine scraper conveyor transmission is a three-stage planetary reducer, mainly bevel gear meshing, helical gear meshing, and planetary reducer; its total transmission ratio is 27, the input speed is 1400 r/min, the power is 1500 KW, the input shaft bevel gear is a faulty gear, and the relevant parameters of HB-KPL-75 gearbox are shown in Table 1.

### 3.2. Current Signal Decomposition of VMD-GA

In this research, the current signal data for a certain period are screened from the collected current signals. As shown in Figure 3, when the coal flow load gradually increases with time, the current signal of the load motor fluctuates, obviously. In addition, the load impact signal causes the current signal to change. Additionally, the current signal is seriously affected by the power frequency signal of the electrical system, which makes it difficult to distinguish effectively the fault types from the time domain. This will seriously affect the effectiveness of feature expression using the Hilbert spectrum and increase the difficulty in fault detection. Therefore, the VMD and GA algorithms are used to decompose the time domain signal first.

GA is used to optimize the parameters of VMD. The first step is to set the number of parameter decomposition layers and the initial range of penalty factor, so as to prevent too small decomposition layers from causing insufficient decomposition of the current signal, and too large decomposition layers from causing excessive decomposition and generating false single-mode component information. After many experiments, the minimum parameter of GA initial decomposition is 3, the maximum parameter is 8, and the penalty factor is 1000~4500. The second step is to set the initialization of GA parameters. The genetic algorithm parameters used in this paper are similar to those in reference [40]: setting the initial population size to 20 can improve the genetic algorithm’s stability and ensure the population’s diversity. Additionally, the crossover rate is 0.70, which can ensure both high fitness structure and search efficiency. Finally, the mutation rate is 0.18. Other GA-related parameter settings are shown in Table 2.

The GA algorithm finds the optimal parameter combination corresponding to the signal. For the four fault forms of gear broken tooth, normal, pitting, and tooth wear, GA iterates to the 4th, 6th, 7th and 4th times respectively to achieve the best fitness, and the fitness values are 0.073, 0.0452, 0.065 and 0.042. The variation in fitness with iteration times in the optimization process is shown in Figure 4.

After the current signal is calculated by the VMD-GA algorithm, the Hilbert spectra of different states are shown in Figure 5a–d, and the right side is the reference coordinate of color depth, which represents the amplitude. The Hilbert spectrogram shows that there is no aliasing and insufficient decomposition in the frequency band, and the current signal is mainly the power frequency signal introduced for the electrical system, which leads to no apparent characteristic information on the Hilbert spectrogram under different fault types. Furthermore, the existence of load shock leads to the larger energy of transient shock in the current signal, manifested as a pulsating shock signal near the fundamental frequency. The normal Hilbert spectrum shown in Figure 5a indicates that the frequency component mainly changes around 500 and 2500 Hz, where the power frequency signal is dominant, and the energy changes obviously with time around 500 Hz. After 0.6 s, the energy increases, and the maximum energy is about 1200. The Hilbert spectrum in Figure 5b shows that the amplitude varies between 0.2~0.4 s, 0.6~0.8, and 0.8~1.0 s with the load increase, and the maximum amplitude is about 100. There is characteristic frequency information between 1 and 200 Hz. For pitting and wear failure modes, the Hilbert spectrum in Figure 5c,d shows that the characteristic information for the dominant energy distribution is similar. They all have abundant characteristic information near 1000 Hz, and the maximum amplitude is 250. It is shown that the Hilbert spectrum of four different fault characteristics are strongly influenced by power frequency signals. The IMF component in different fault signals contains the common information for normal signals and fault signals, and the information unrelated to the fault should be removed.

### 3.3. Sensitivity Calculation of Fault Sub-Band

After decomposing three kinds of fault signals by VMD-GA, six IMF components are obtained from broken tooth signals, and five IMF components are obtained from gear wear and pitting signals. According to the sensitive IMF discrimination algorithm, the correlation coefficient of IMF components for each fault type is calculated $\left[\alpha_{i},\beta_{i},\gamma_{i},\lambda_{i},\max(d_{n})\right]$, as shown in Figure 6.

The IMF1, IMF2, and IMF3 of the broken tooth signal; IMF1, IMF2, IMF3, and IMF4 of the pitting signal; and IMF1, IMF2, and IMF3 of the wear signal are selected by the sensitive IMF discrimination algorithm, and the results are shown in Figure 7.

As shown in Figure 7a–c, the local instantaneous energy spectrum of the broken tooth fault signal has prominent impact characteristics compared with the normal gear meshing signal. In contrast to pitting and tooth wear, the broken tooth condition has abundant characteristic information at 0–200 Hz, and the highest amplitude is 40. Compared with wear, pitting has characteristic information at 800–1000 Hz, and the characteristic frequency information is narrower than the wear frequency range, especially at 0–0.3 s and 0.7–1 s, and the maximum amplitude is 30. The amplitude of pitting near 500 Hz is higher than that for gear wear. Therefore, the distribution of the Hilbert spectrum in different states is different, and the feature dataset for fault identification can be constructed.

Then, using the same process, 1200 characteristic datasets are generated by VMD-GA combined with the Hilbert spectrum for current signals under normal, broken teeth, pitting, and wear conditions. Part of the dataset is shown in Figure 8.

### 3.4. Feature Learning and Pattern Classification Based on ShuffleNet-V2

ShuffleNet-V2 network architecture can be applied to diagnose gearbox fault types under uneven load shock conditions. The ShuffleNet-V2 neural network is a network with very low complexity. The core of ShuffleNet-V2 consists of two operations: point-by-point group convolution and channel shuffling. Point-by-point group convolution can significantly reduce computational loss. At the same time, channel shuffling can effectively alleviate the insufficient information flow between point-by-point group convolution channel groups, thus realizing the most advanced performance [41,42], as shown in Figure 9a. In ShuffleNet-V2 unit 1, the input feature graph is divided into two branches, and the number of channels each account for 1/2. The lower branch is constant; the upper branch goes through three convolutions with a step size of 1, using the same number of input and output channels. Two 1 × 1 convolutions are ordinary, and a 3 × 3 convolution is deep convolution in deep separable convolution. When the convolution is completed, the two branches carry out the Concat operation, add the number of channels, fuse features, and finally use Channel Shuffle to exchange information between different groups, so that the channels can be fully fused. In ShuffleNet-V2 Unit 2, instead of channel partitioning, the feature map is directly input to both branches. Both branches use 3 × 3 depth convolution with a step size of 2 to reduce the dimension of the feature map’s length (H) and width (W), thus reducing the network computation. Then, the Concat operation is carried out after the output of the two branches, and the number of channels added is twice that of the original input, increasing the network’s width and giving the network stronger feature extraction ability. Finally, channel shuffling is also carried out to realize the information exchange between different channels, as shown in Figure 9b.

The overall network structure of ShuffleNet-V2 is shown in Figure 9. The input feature map size is 3 × 224 × 224. First, 24 3 × 3 ordinary convolutions with a step size of 2 are used for feature extraction, and then the maximum pooling layer is used for down sampling. Then, three module layers composed of ShuffleNet-V2 Unit 2 and ShuffleNetV2-Unit 1 are used continuously, and the number of units 2 and 1 in the module layer is 1:3, 1:7, and 1:3, respectively. Then, 1024 1 × 1 convolution channels with a step size of 1 are used to expand the number of channels, and different characteristic information for the fault gears is obtained through the large receptive field of the large channel and large convolution.

In the neural network training process, 1200 datasets of four states are established, and 300 datasets are established for each working condition, of which 70% are training sets, 30% are testing verification sets, and 30 testing sets are reserved for each working condition. Moreover, hyperparameters have a certain influence on network performance, including the learning rate and optimizer. The unreasonable setting of the learning rate will cause the loss value to be challenging to find the decreasing direction quickly, leading to an increase in training time and the loss value remaining unchanged. On the contrary, if the learning setting is too large, there will be an overshoot. Through a large number of experiments, the learning rate of this selection is 0.001, the optimizer is selected as Adam, and the L2 regularization factor is set to 4 to prevent overfitting. The training correlation results are shown in Figure 10 as the training correlation curve.

As shown in Figure 10, the cross-entropy loss and accuracy of the verification and training set change with the increase in iteration times. The double longitudinal axes represent the accuracy and cross-entropy loss information, respectively. With the increase in training iteration times, the cross-entropy loss decreases rapidly when epoch = 20, then falls and finally approaches 0 gradually, indicating that the model can learn features from the training set. After 120 iterations, the cross-entropy loss and accuracy of the model converge to stable values. Finally, the accuracy of the model verification is also continuously improved to 94.79%, while the accuracy of the test set is 91.66%. Furthermore, these results show that the depth model has excellent performance and good fault diagnosis ability.

### 3.5. Contrast Experiment

To evaluate the performance of the proposed method in gear fault diagnosis of the scraper conveyor gearbox, a comparative test is carried out on the same current dataset, and the characteristic dataset is generated by a wavelet time-frequency transform. Through many experiments, the wavelet basis of wavelet time-frequency transform is selected as Db45, and some results are shown in Figure 11. Through the high-frequency part of the wavelet time-frequency map, we can find that wavelet time-frequency images contain abundant time-frequency characteristics of fault features [43]. In addition, two different datasets are used to train the RestNet-18 neural network.

The related training process and information are shown in Table 3.

As shown in Table 3, the average accuracy of the dataset generated by ShuffleNet-V2 combined with a wavelet time-frequency map is lower than that for the method proposed in this paper. In addition, the process of generating the wavelet time-frequency map datasets requires monitors to conduct many experiments to match the best wavelet basis. By testing the network after training, the network proposed in this article has the highest test accuracy, but compared with ShufferNet-V2, RestNet-18 training takes less time.

## 4. Result and Analysis

There are two possible reasons behind the performance enhancement of the model proposed in this paper. The first reason is that a suitable feature expression method is introduced into the gearbox fault diagnosis model. Even under the working conditions of load shock and power frequency signal interference, the proposed diagnosis model can extract unique fault feature information from input signals of different health conditions without manual intervention. Therefore, when the training dataset input is provided to the classifier (i.e., ShuffleNet-V2), it can accurately classify the data. The second reason is that the ShuffleNet-V2 neural network is used in the classification stage. Because of its simple and efficient network structure, it achieved higher accuracy after a short time.

## 5. Conclusions

In this paper, the current signal is used as the input data in the fault diagnosis model, which can effectively avoid the shortcomings caused by the vibration signal, and the fault diagnosis for the gear in the mine scraper conveyor gearbox is realized by MCSA. The results of this study are as follows:

1. Fault diagnosis of the gearbox gear in the mine scraper conveyor is always a difficult problem in related fields. The Hilbert spectrum is used to express the fault signal’s characteristics directly. The redundant irrelevant information negatively influences the characteristic fault information, mainly the power frequency and load impact signals caused by the electrical system. In this paper, the fault-sensitive IMF signal is used for extraction, and the experimental results prove the effectiveness of this method.

2. The higher diagnostic accuracy also shows that different types of fault gears have a certain degree of influence on the time, frequency, and amplitude of the load motor’s current signal. Taking it as an index, it can overcome the fault diagnosis problem under the condition of an unknown fault characteristic waveform.

3. For calculating fault-sensitive parameters of the scraper conveyor, the fault information-sensitive sub-band should be in the position of IMF1, IMF2, IMF3, and IMF4, and the IMF5 component is mainly similar to the normal signal waveform characteristics.

4. We prove that the feature extraction method based on VMD-Hilbert spectrum–ShuffleNet-V2 achieved high accuracy quickly, and it is a more suitable fault diagnosis model for a mine scraper.

Finally, this article has some shortcomings because the proposed data preprocessing methods have ample space for improvement in the implementation complexity. In addition, there is still room for further research on the performance and interpretability of the ShuffleNet-V2 model.

## Figures and Tables

**Figure 1 sensors-23-04951-f001:**
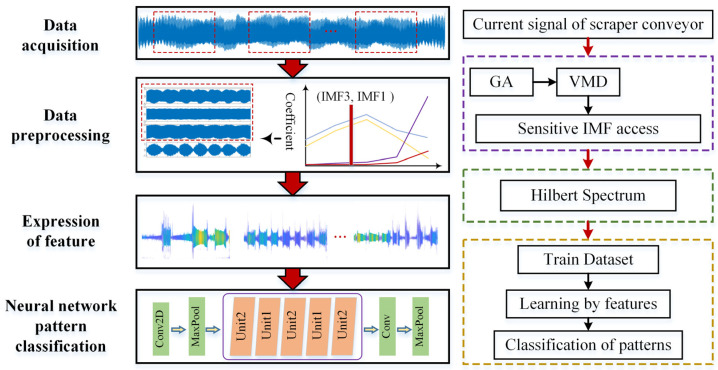
Flow chart showing fault diagnosis for the scraper conveyor gearbox.

**Figure 2 sensors-23-04951-f002:**
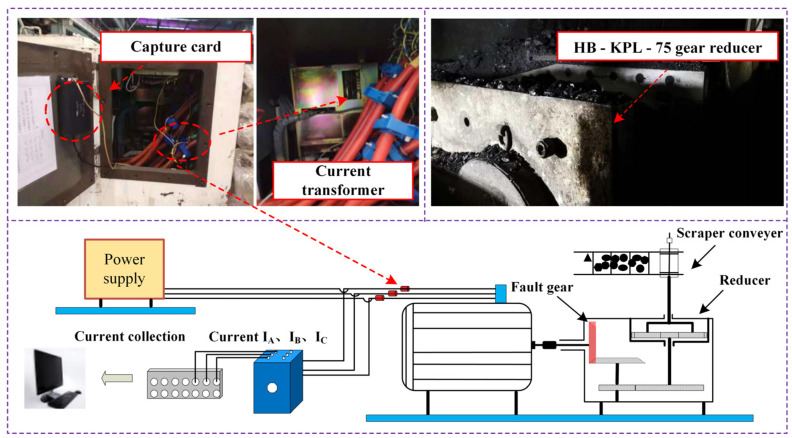
HB-KPL-75 reducer and data acquisition.

**Figure 3 sensors-23-04951-f003:**
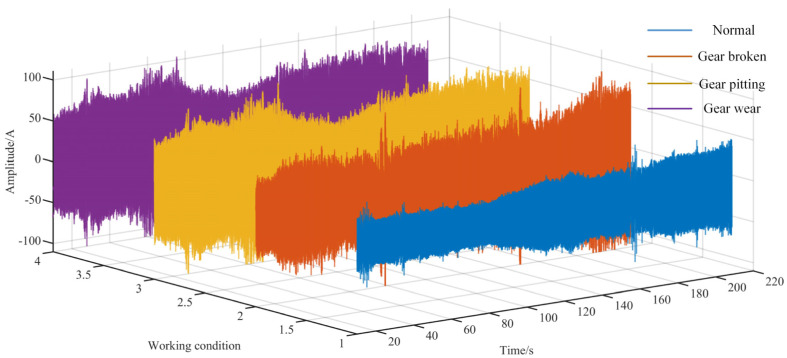
Reducer current signal in different states.

**Figure 4 sensors-23-04951-f004:**
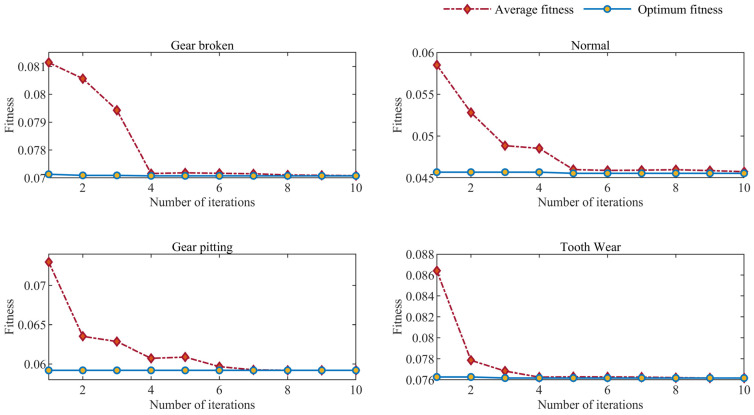
Variation in fitness–iteration times.

**Figure 5 sensors-23-04951-f005:**
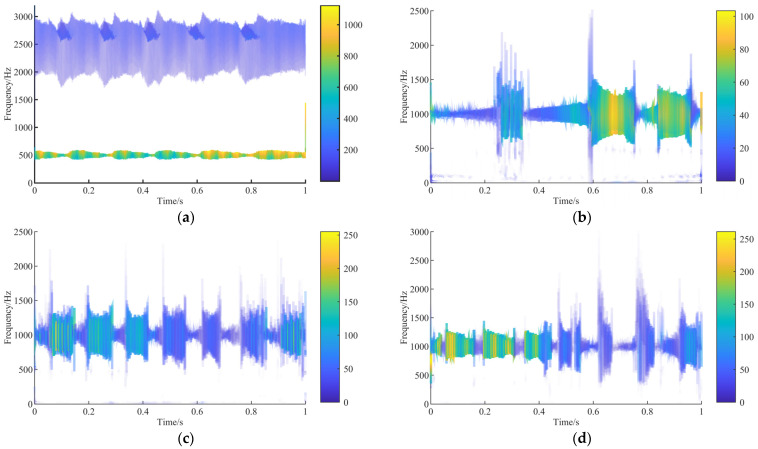
Decomposition of raw data by VMD-GA: (**a**) normal; (**b**) gear broken; (**c**) gear pitting; (**d**) tooth wear.

**Figure 6 sensors-23-04951-f006:**
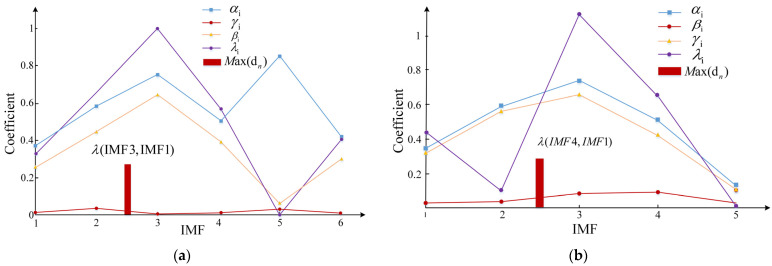
Sensitive IMF discrimination results for different fault signals: (**a**) gear broken; (**b**) gear pitting; (**c**) tooth wear.

**Figure 7 sensors-23-04951-f007:**
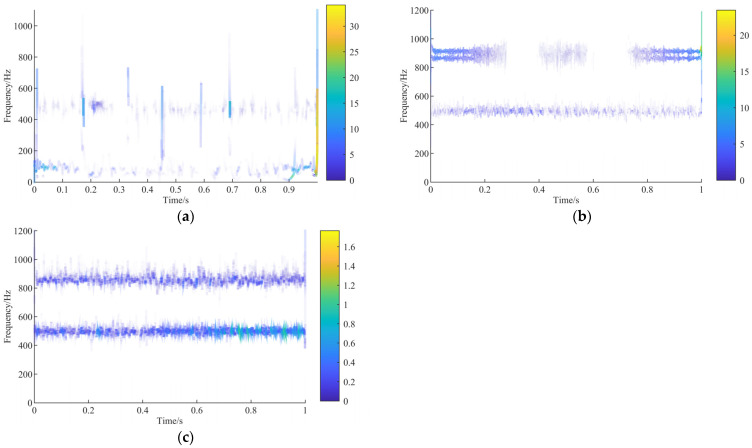
Excluding irrelevant components in the Hilbert spectrum: (**a**) gear broken; (**b**) gear pitting; (**c**) tooth wear.

**Figure 8 sensors-23-04951-f008:**
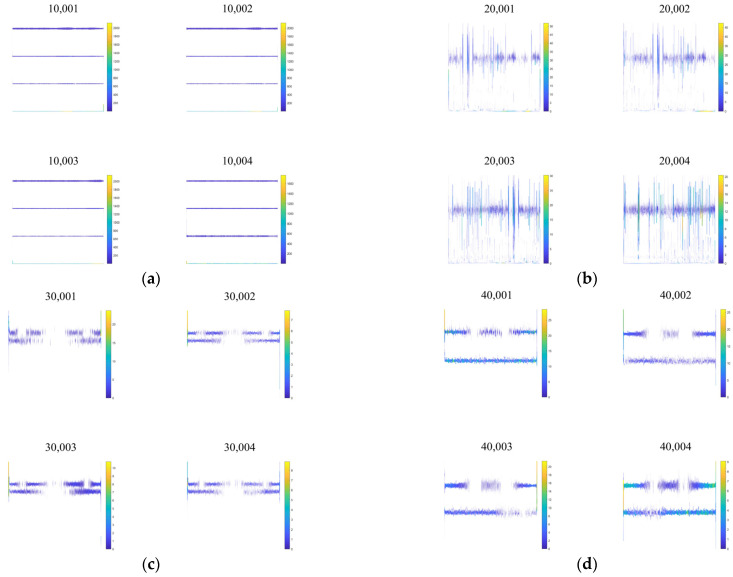
Hilbert spectrum dataset for different fault types: (**a**) normal; (**b**) gear broken; (**c**) gear pitting; (**d**) tooth wear.

**Figure 9 sensors-23-04951-f009:**
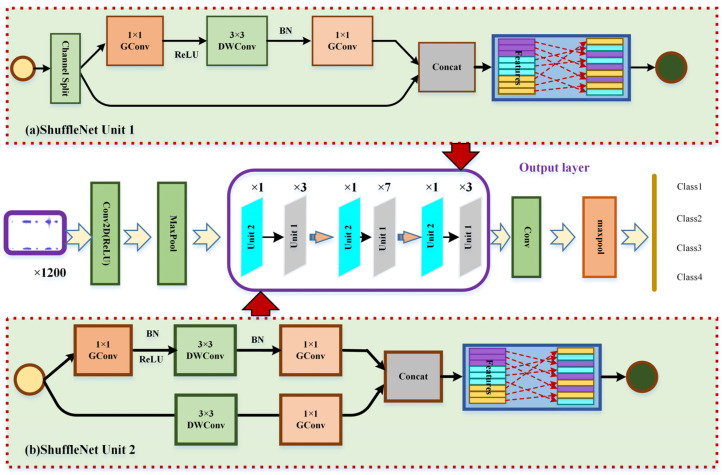
ShuffleNet-V2 model.

**Figure 10 sensors-23-04951-f010:**
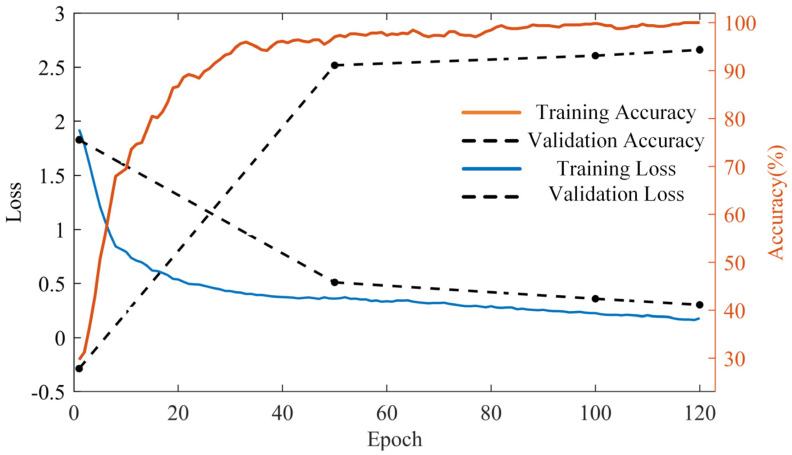
ShuffleNet-V2 network training curve based on the Hilbert spectral dataset.

**Figure 11 sensors-23-04951-f011:**
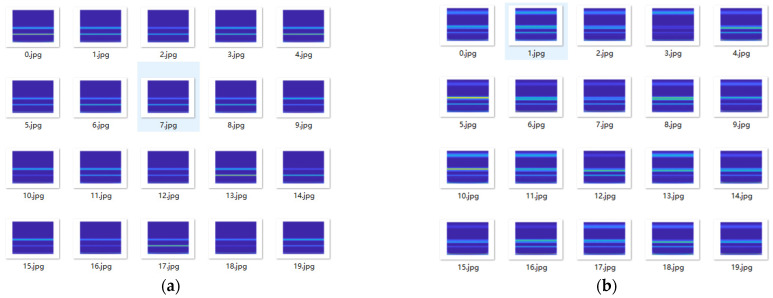
Wavelet decomposition time–frequency analysis dataset: (**a**) normal; (**b**) gear broken; (**c**) gear pitting; (**d**) tooth wear.

**Table 1 sensors-23-04951-t001:** HB-KPL-75 gearbox related parameters.

Gear Name	Number of Teeth
Bevel gear (driving)	36
Bevel gear (driven)	70
Helical gear (driving)	36
Helical gear (driven)	83
Sun wheel	17
Planetary gear	21
Planetary gear	71

**Table 2 sensors-23-04951-t002:** Parameter settings for the genetic algorithm.

Parameter	Setting
Penalty factor	1000~4500
Decomposition layers	3~8
Generations	10
Population size	20
Mutation probability	0.18
Crossover probability	0.7

**Table 3 sensors-23-04951-t003:** Training process information for different datasets.

Neural Network	Dataset	Duration (/s)	Hardware Resources	Learning Rate	Training Accuracy Rate (%)	Validation Accuracy Rate (%)
ShuffleNet-V2	VMD-GA-Hilbert spectrogram	778	GTX1660ti	0.001	94.35	91.66
ShuffleNet-V2	Wavelet time–frequency	780	GTX1660ti	0.001	91.13	90.00
ResNet-18	VMD-GA-Hilbert spectrogram	144	GTX1660ti	0.0001	85.00	84.17
ResNet-18	Wavelet time–frequency	185	GTX1660ti	0.0001	83.60	80.83

## Data Availability

The data presented in this study are available on request from the corresponding author.

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
