# Peer review of "Intelligent Fault Diagnosis Method Based on VMD-Hilbert Spectrum and ShuffleNet-V2: Application to the Gears in a Mine Scraper Conveyor Gearbox"

_sensors, 2023, doi:10.3390/s23104951_

Round 1
Reviewer 1 Report
All my comments on the article are in the attached file.

The quality of English language of the article had better be improved.
Reviewer 2 Report
This paper proposes a fault diagnosis methodology for mine scraper conveyor gearbox gears using motor current signature analysis, variational mode decomposition and ShuffleNet-V2.
- The introduction motivates the research, but several papers relevant to the topic and inherent to machine learning and deep learning are missing.
- The method should be compared with other methods in the literature. Also, ResNet-50 may be too large for the number of data available; a smaller residual network, e.g., resnet-18, should also be evaluated. VGG networks could be considered as well.
- A table should be included where the number of data contained in the training sets and test sets is clearly made explicit.
- Reported accuracies should be described in training and test. Any overfitting problems should be commented on.
- The learning reate, optimizer and network regularization should be better presented explained and commented.
- Figure 10 shows the test loss and test accuracy. Since these metrics are represented for each epoch, it is clear that what the authors state as a test set is actually a validation set. The paper should be revised in accordance with this observation. Furthermore, the methodology should be tested on a real test set and not a validation set. Test data should never be used for training.
- Line 70: Fan et al. the reference is missing.
- Line 380: it is not clear how Figure 8 shows 600 characteristic data sets. Please, revise.
- The quality and readability of all images needs to be improved. For example, some images related to Hilbert spectrum are illegible. For example, Figure 8, 11 and 12. In some images it is evident that one is selected from a screenshot.
Language should be revised.
Language should be revised.
Reviewer 3 Report
In the article, the authors presented the diagnosis of gear damage on mine scraper conveyors using motor current signature analysis (MCSA), which solves the problem that information about the gear damage characteristics is influenced by information about coal flow load and power frequency, leading to the problem that damage features are difficult to extract effectively.
The success of the authors is the proposed method of diagnosing damage based on variation mode decomposition (VMD)-Hilbert spectrum and ShuffleNet-V2.
It is interesting to use the gear current signal is decomposed into a series of internal mode functions (IMF) using VMD, and sensitive VMD parameters are VMD parameters are optimized using a genetic algorithm (GA). The sensitive IMF algorithm is used to evaluate the modal function sensitive to error information after VMD processing.
The analysis of research and the accuracy of the obtained algorithms leads to the right conclusions. However, the conclusions should take into account the directions of further research and their possibilities in the future.
The article is well written and can interest the readers and contribute to the uplift
work safety and proper use of the mining scraper conveyor.
Round 2
Reviewer 2 Report
The quality of the figures should be improved. For instance, figure 8 does not convey anything. Only a few examples could be shown but with better quality.
Minor editing of English language required
